# Electrochemical halogen-atom transfer alkylation via α-aminoalkyl radical activation of alkyl iodides

Xiang Sun [1] & Ke Zheng [1] ✉

Alkyl halides, widely recognized as important building blocks and reagents in organic synthesis, can serve as versatile alkyl radical precursors in radical-based transformations. However, generating alkyl radicals directly from unactivated alkyl halides under mild conditions remains a challenge due to their extremely low reduction potentials. To address this issue, α-aminoalkyl radicals were employed as efficient halogen-atom transfer (XAT) reagents in the photoredox activation of unactivated alkyl halides. Here, we report an effective electrooxidation strategy for generating alkyl radicals from unactivated alkyl iodides via an electrochemical halogen-atom transfer (e-XAT) process under mild conditions. The α-aminoalkyl radicals generated by anodic oxidation are demonstrated to be efficient XAT reagents in these transformations. This facile electricity-driven strategy obviates the need for sacrificial anodes and external chemical oxidants. The method successfully applies to a wide variety of alkyl iodides, including primary, secondary, and tertiary, as well as structurally diverse olefins, exhibiting excellent functional group tolerance. Moreover, we further demonstrate the utility of this strategy by rapidly functionalizing complex molecules and biomolecules.

Alkyl halides are essential building blocks and reagents in organic synthesis and drug discovery, given their structural diversity and synthetic availability. Recently, they have been used as alkyl radical precursors in radical-based transformations[1–6]. However, accessing alkyl radicals from unactivated alkyl halides has proven challenging due to their extremely low reduction potentials ($E_{red} < -2.0$ V $vs$ SCE). Currently, methods for accessing alkyl radicals can be primarily categorized into two strategies based on mechanisms: the single-electron transfer (SET) strategy and the halogen-atom transfer (XAT) strategy. The former has proven highly reliable in transition-metal catalysis[7–15] and photoredox systems[16–19]. Fu and Peters demonstrated that the excited-state Cu-nucleophile species could reduce alkyl halides to generated alkyl radicals via the SET process[20,21]. Notably, significant progress has been achieved by the development of using tin and silicon reagents as well as trialkylborane–$O_2$ systems, which efficiently generate alkyl radicals via the halogen-atom transfer (XAT) process

(Fig. 1a)[22–31]. For example, MacMillan recently demonstrated that transient silicon radicals could assist in generating alkyl radicals from alkyl halides via photoredox activation by metallaphotoredox catalysts[32–36]. Most recently, Leonori and Juliás developed an efficient strategy using strongly nucleophilic α-aminoalkyl radicals as halogen-atom transfer (XAT) agents, which can be easily accessed from simple amines under photoredox systems. These α-aminoalkyl radicals can promote the homolytic activation of carbon-halogen bonds to produce alkyl radicals under mild conditions[37–42]. However, these strategies uniformly rely on expensive photocatalysts (PC) in generating the halogen-atom transfer agents. Therefore, simpler, milder, and more efficient strategies for activating unactivated alkyl halides remain highly desired.

Over the past decade, organic electrochemistry has garnered increasing attention due to its finely-tuned electron-transfer processes and its utilization of electrons as traceless redox reagents[43–51]. One such

[1]Key Laboratory of Green Chemistry & Technology, Ministry of Education, College of Chemistry, Sichuan University, Chengdu 610064, PR China.
✉e-mail: kzheng@scu.edu.cn

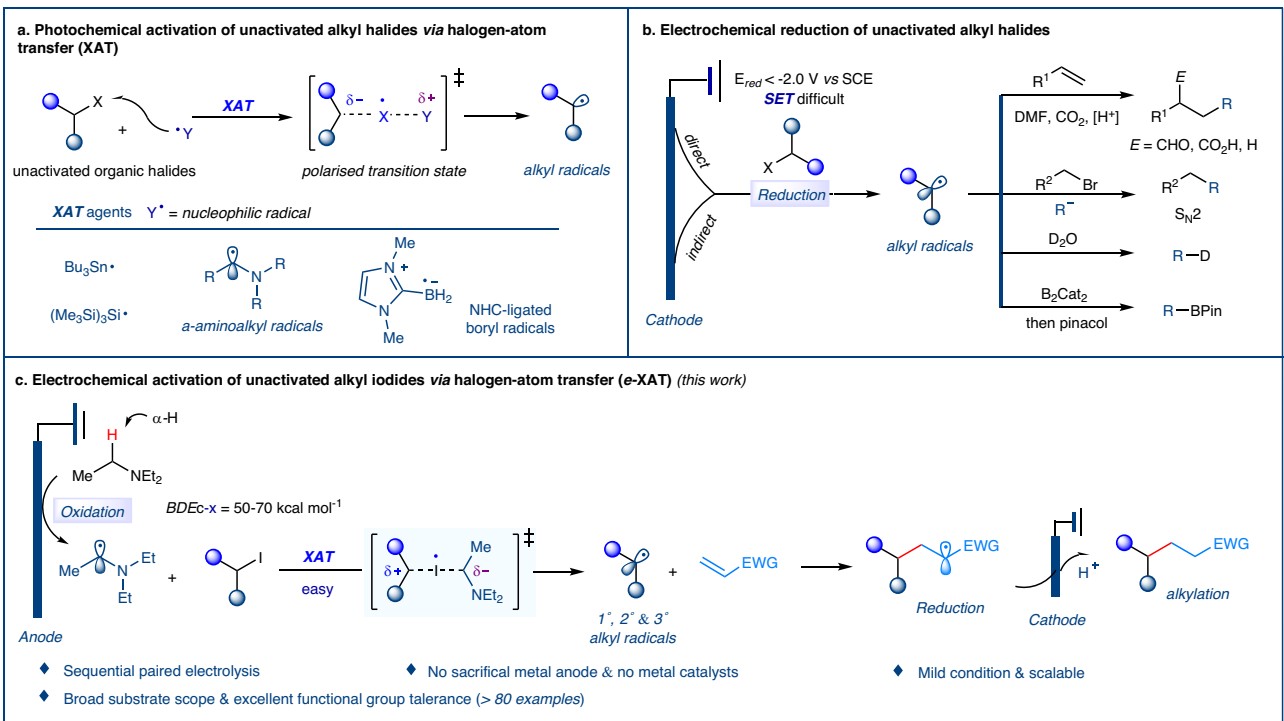

**Fig. 1 | Methods for activation of unactivated alkyl halides. a** Photochemical activation of unactivated alkyl halides via halogen-atom transfer (XAT). **b** Electrochemical reduction of unactivated alkyl halides. **c** Electrochemical activation of unactivated alkyl iodides via halogen-atom transfer (e-XAT).

process, electroreduction, has demonstrated its ability to produce alkyl radicals through the cathodic reduction of unactivated alkyl halides. This concept has been elegantly explored in a plethora of contributions studying the electroreduction of alkyl halides for hydrodehalogenation, carboxylation, and alkylation of alkenes[52–59]. However, these reactions have typically been hindered by their reliance on sacrificial anodes, which necessitate a large excess of alkenes or alkyl halides, a narrow substrate scope, and low efficiency and/or chemoselectivity. Recently, significant progress has been made in electroreductive carbofunctionalization of alkenes with alkyl bromides, cross-electrophile coupling (e-XEC), borylation, and deuteration of alkyl halides (Fig. 1b)[60–63]. In general, unactivated alkyl halides with reduction potentials below −2.0 V vs SCE are predominantly reduced at the cathode using inert electrode materials such as carbon or platinum. To access higher reduction potentials at the cathode, anode materials such as magnesium, nickel, or zinc are commonly employed, but are consumed during the electrolysis process. Therefore, these electroreduction methods typically require the use of sacrificial anodes[60–62]. Among them, the electroreductive dehalogenation deuteration stands as an exception, as it does not require a sacrificial anode due to the utilization of high current (30 or 500 mA)[63]. This has raised a fundamental question: can alkyl radicals be accessed through an electrooxidation strategy, which would allow for the recycling of electrode materials and expansion of the substrate scope.

In this study, we present a novel and versatile electrooxidation method for the generation of α-aminoalkyl radicals, which act as halogen-atom transfer agents for the activation of unactivated alkyl iodides. This mild and general electrochemical approach has been successfully applied to the hydroalkylation of electron-poor olefins (Fig. 1c). The proposed method offers a new means for activating unactivated alkyl iodides to produce alkyl radicals, without the use of a sacrificial anode. This method exhibits a broad substrate scope, as demonstrated by the successful conversion of a diverse range of alkyl iodides (including primary, secondary, and tertiary substrates) and structurally varied olefins into alkylation products with excellent functional group tolerance. Moreover, we demonstrate the utility of

this electrooxidation strategy through the rapid functionalization of complex molecules. Overall, this method provides a valuable contribution to the field of organic electrochemical synthesis and offers a powerful tool for accessing a wide range of alkyl radicals under mild conditions.

## Results and discussion
### Reaction optimization
We conducted an investigation of the electrochemical halogen-atom transfer (e-XAT) alkylation reaction using 3-iodo-*N*-Boc-azetidine **1** and acrylonitrile **2** as the model substrates (Table 1). For reaction optimization, we selected Et₃N (Et₃N: $E_{ox}$ = +0.77 V vs SCE)[37] as the XAT-agent precursor and H₂O as the H-atom donor. After exploring various electrolytes, amines, electrode materials, and solvents (see Supplementary Information for details), we identified the optimal conditions as constant current electrolysis (3 mA) in CH₃CN/H₂O (10:1) at 50 °C with Et₃N as the XAT-agent precursor in an undivided cell (a Schlenk tube) using a simple two-electrode configuration (RVC anode and Pt plate cathode). The desired alkylation product **3** was obtained in 86% yield under these conditions (entry 1). Control experiments confirmed the importance of electricity (entry 2), XAT-agent precursor (entry 3), and undivided cell (entry 4) for the transformation, while no reactions were observed in the absence of Et₃N, electricity, and electrolysis in a divided cell, respectively. The use of other electrolytes led to decreased yields, with Et₄NPF₆ giving only 16% yield (entries 5–7). Reducing the amount of H₂O resulted in a slightly lower yield (entry 8). In addition, increasing or decreasing the current density led to a noticeable drop in reaction efficiency (entries 9–10). Other electrode materials, such as Mg cathode and Zn cathode, were found to be less effective (entries 11–12).

### Reaction scope
We established the optimal conditions and proceeded to investigate the generality of the e-XAT process with respect to various Michael acceptors (Fig. 2). High to excellent yields of the corresponding alkylation products were efficiently obtained from a diverse range of

**Table 1 | Optimization of the reaction conditions[a]**

| Entry | Variation from the standard conditions | Yield [%][b] |
|---|---|---|
| 1 | None | 86[c] |
| 2 | No electricity | 0 |
| 3 | No Et$_3$N | 0 |
| 4 | Divided cell | 0 |
| 5 | $n$-Bu$_4$NBF$_4$, 24 h, rt | 70 |
| 6 | Et$_4$NPF$_6$, H$_2$O (0.40 mL), rt | 16 |
| 7 | $n$-Bu$_4$NPF$_6$, H$_2$O (0.40 mL), rt | 61 |
| 8 | H$_2$O (0.40 mL), rt | 73 |
| 9 | 2.0 mA, rt | 52 |
| 10 | 4.0 mA, rt | 60 |
| 11 | H$_2$O (0.40 mL), RVC(+)-Mg(-), rt | 15 |
| 12 | RVC(+)-Zn(−), rt | 41 |

[a]Undivided cell, RVC anode (100 PPI), Pt plate cathode, **1** (0.3 mmol), **2** (0.6 mmol), CH$_3$CN (4.5 mL), H$_2$O (0.45 mL), N$_2$, 3.0 mA, 10 F/mol, 28 h.
[b]Yield determined by $^1$H-NMR analysis using 1,3,5-trimethoxybenzene as the internal standard.
[c]Isolated yield.

electron-poor olefins. Coupling partners with different functional groups, including nitrile, ester, carboxylic acid, sulfone, ketone, phosphonate, amide, and pyridine (**3**–**11**), were shown to be suitable. Acrylates containing reactive functional groups such as CF$_3$, halides (Cl, Br), and epoxide (**12**–**15**), were well-tolerated under these electrooxidation conditions. Furthermore, moderate to high yields of alkylation products were obtained from sterically hindered α- or β-alkyl acrylates, lactone, and cyclic ketones (**16**–**22**).

Unnatural amino acids and their derivatives play a crucial role in the synthesis of many pharmaceuticals, bioactive compounds, and agrochemicals. A reliable protocol for convenient access to structurally complex unnatural amino acids is highly desirable. In this context, we have explored the scope of the *e*-XAT strategy to synthesize diverse unnatural amino acids from Boc-protected dehydroalanine and various alkyl iodides (**23**–**49**). This approach successfully converted a range of secondary N-heterocyclic alkyl iodide derivatives, including azetidine, pyrrolidine, piperidine, azepane, nortropine, proline, and 2-aminopyrimidine, to the corresponding products in good to excellent yields (**23**–**31**). We also found that cyclic alkyl iodides with different ring sizes, such as five-, six-, seven-, and 12-membered frameworks, could be conveniently converted to the desired alkylation products (**32**–**36**) in yields ranging from 54% to 96%. Additionally, we successfully transformed 4-iodo(thio)pyrans, 3-iodooxetane, and cyclobutene derivatives into the desired products in moderate to good yields (54–96%, **37**–**40**). Remarkably, alkyl iodides with various functional groups, such as acetal, ether, ketone, amide, and N-Boc amine, were well-tolerated in this *e*-XAT method, delivering the desired products in high yields (67–95%, **41**–**45**). Spirocyclic and bicyclic derivatives, which are important structural features in drug development due to their high C(sp$^3$) content, were also found to be suitable substrates (62–97% yields, **46**–**49**).

To further evaluate the generality of this *e*-XAT protocol, we next investigated the efficacy of the process on primary and tertiary alkyl iodides. Primary alkyl iodides with varying carbon chain lengths (**51**–**52**) yielded the desired products in high yields (72–97%). A wide range of functional groups, including chloride, terminal alkynes, terminal alkenes, hydroxyl, TBS-protected ether, nitrile, silane, ketone, ester, and perfluoroalkyl (**53**–**62**), were found to be compatible under these mild electrochemical conditions. These functional groups represent useful synthetic handles and provide excellent opportunities for further synthetic manipulation. As an example of the application of the protocol to diiodides, 1,3-diiodopropane delivered the desired product **63** in 72% yield. Additionally, cyclic and bicyclic tertiary alkyl iodides (**65**–**70**) were demonstrated to be suitable substrates.

The exceptional functional group compatibility of this electrochemical strategy inspired us to explore its potential for the functionalization of structurally complex molecules. As illustrated in Fig. 3, a range of acrylates (**71**–**76**) derived from natural products including Nopol, Citronellol, Terpineol, Geraniol, Cedrol, and (+)-Fenchol were compatible with this facile protocol and yielded the corresponding products in high yields. Moreover, various primary and secondary alkyl iodides derived from Nopol (**77**), Citronellol (**78**) and biomolecules (ribofuranoside **79**, xylofuranose **80**, glucofuranose **81**, galactopyranose **82**, fructopyranose **83**) were effectively converted to versatile alkyl radicals via this *e*-XAT process. These alkyl radicals subsequently reacted with Boc-protected dehydroalanine to form a variety of structurally complex unnatural amino acid derivatives in moderate to high yields (**77**–**83**). Notably, alkyl iodides derived from the pharmaceutical agent Probenecid (**84**) delivered the desired alkylation product in 72% yield. Furthermore, the synthetic utility of this method was demonstrated by gram-scale synthesis, yielding the alkylation products in high yields (Fig. 4e).

## Mechanistic investigations

In order to rationalize the *e*-XAT strategy, cyclic voltammetry studies were conducted. These studies revealed that triethylamine (Et$_3$N) underwent oxidation at a significantly lower potential compared to the reduction of alkyl iodides and electron-poor olefins (Fig. 4d). Previous studies have shown that α-aminoalkyl radicals can activate alkyl

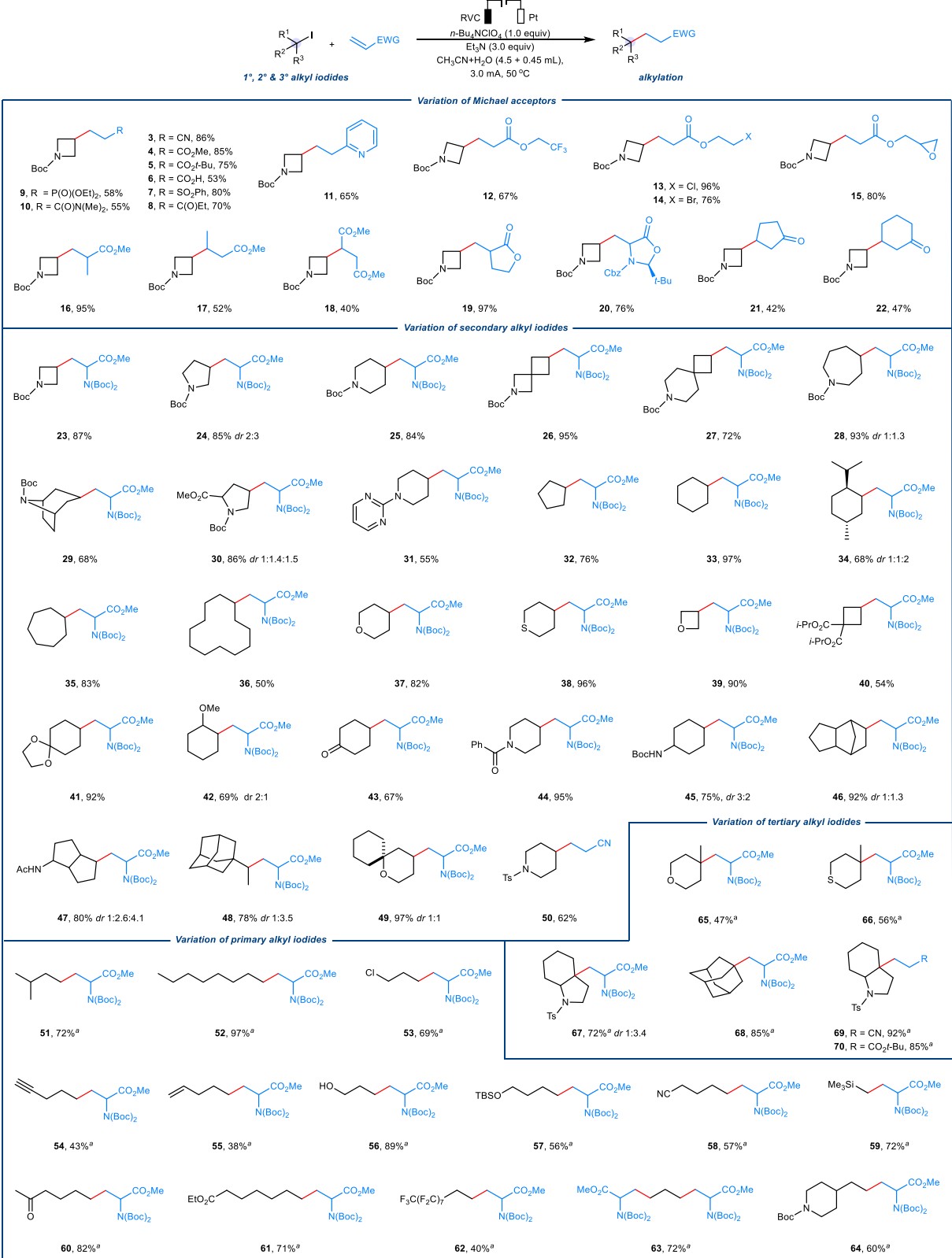

**Fig. 2 | Scope of the electrochemical halogen-atom transfer (*e*-XAT) alkylation.** Reaction conditions: RVC anode, Pt cathode, alkyl iodides (0.3 mmol), CH₃CN (4.5 mL), H₂O (0.45 mL), N₂, 3.0 mA, 10 F/mol, 50 °C, 28 h. $^a$48 h, 18 F/mol. Isolated yields.

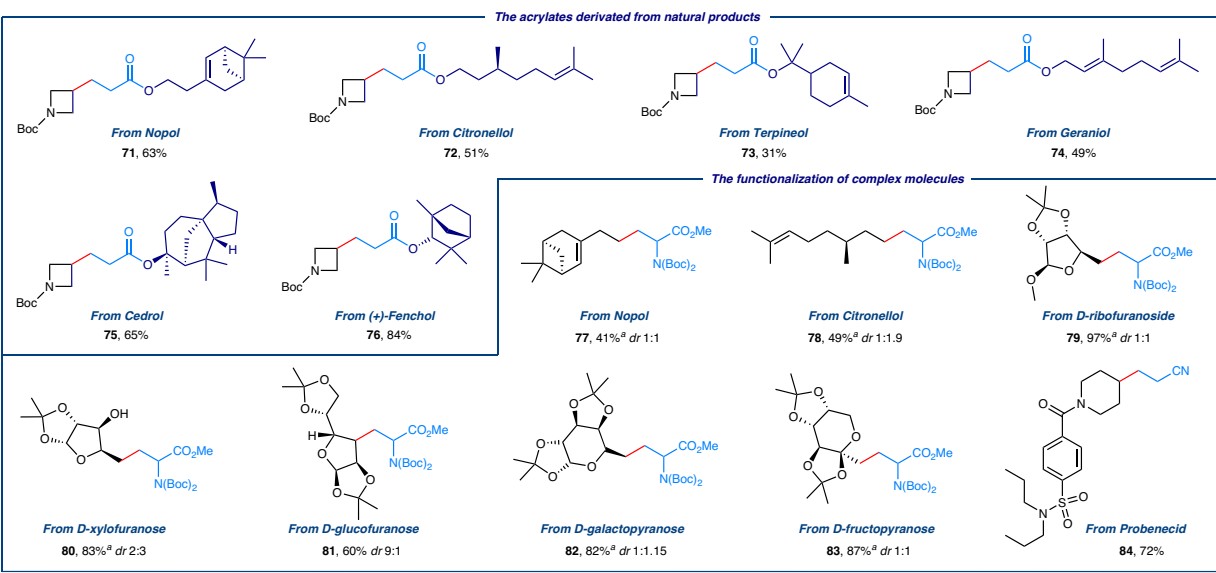

**Fig. 3 | Functionalization of complex molecules via electrochemical halogen-atom transfer (*e*-XAT).** Reaction conditions: RVC anode, Pt cathode, alkyl iodides (0.3 mmol), CH$_3$CN (4.5 mL), H$_2$O (0.45 mL), N$_2$, 3.0 mA, 10 F/mol, 50 °C, 28 h. [a]48 h, 18 F/mol. Isolated yields.

iodides via an iodine-atom transfer interplay of polar effects, leading to the generation of alkyl radicals[37]. Based on these observations, we speculated that the electrochemical protocol might follow a radical-polar crossover pathway initiated through the generation of α-aminoalkyl radicals via anodic oxidation, followed by subsequent deprotonation. The highly nucleophilic α-aminoalkyl radicals may act as halogen-atom transfer agents in the reaction, promoting the homolytic activation of carbon-halogen bonds to form corresponding alkyl radicals. Density functional theory calculations predicted that the XAT process could be kinetically feasible through a polarized transition state with a notable charge-transfer character[37].

A series of control experiments were conducted to support this speculation. Firstly, various amines were tested as XAT-agent precursors under standard electrochemical conditions. The results showed that alkylamines play a fundamental role in this transformation (Fig. 4a). Alkylamines with suitable oxidation potential and secondary α-H (e.g., Et$_3$N and *i*-Pr$_2$NEt) gave better outcomes compared to other electron donors. The use of 2,2,6,6-tetramethyl-N-methylpiperidine with primary α-hydrogens and *i*-Pr$_2$NH with tertiary α-hydrogens resulted in a noticeable reduction in reaction efficiency. These outcomes might be attributed to the primary or tertiary α-aminoalkyl radicals were more difficult to generate during the initiating stage (electrochemical oxidation stage), while secondary α-aminoalkyl radicals demonstrated better stability (compared to primary α-aminoalkyl radicals) and encountered less steric hindrance (compared to tertiary α-aminoalkyl radicals) in the XAT step. The replacement of Et$_3$N with alkylamines without *a*-H (those unable to generate α-aminoalkyl radicals), such as Ph$_2$N(PMP) or 2,2,6,6-tetramethylpiperidine, completely suppressed the reactivity. Similarly, tribenzylamine was unable to initiate the reaction due to its high oxidation potential (E$_{ox}$ = +1.18 V vs SCE). When the radical probe substrate **85** was subjected to standard conditions, the cyclic product **86** was obtained in moderate yield with no acyclic product detected (Fig. 4b, top). This was likely due to a sequence of halogen-atom transfers, intramolecular alkyl radical additions, and iodide radical trapping. Furthermore, the addition of electron-poor olefins to the reaction resulted in moderate yields of the alkylation products **87a** and **87b**, along with **86** (Fig. 4b, middle). These results provided direct evidence that the reaction underwent a radical-polar crossover pathway. Additional evidence was obtained from a free radical clock

experiment using cyclopropyl-containing alkyl iodide **88** as a substrate, which produced the ring-opened product **89** in 36% yield with 12% yield of unrearranged product **90** (Fig. 4b, bottom). This indicated that the rate of intermolecular radical trapping by electron-poor olefins was very fast. The use of D$_2$O instead of H$_2$O under standard conditions yielded the deuterated product **91** in 89% yield, suggesting that the carbanion intermediates were generated by cathode reduction and that H$_2$O acted as a H-atom source in this electrochemical process (Fig. 4c). All of these observations indicated that the activation of alkyl iodide via *e*-XAT was the fundamental process in generating alkyl radicals in this electrochemical alkylation reaction, and that the reaction proceeded through a radical pathway.

Based on the above mechanistic results and the previous reports[37,64–68], we propose a mechanism for the *e*-XAT strategy illustrated in Fig. 4f. Initially, under standard electrochemical conditions, alkylamine Et$_3$N undergoes anodic oxidation, and deprotonation to generate α-aminoalkyl radical **I**, which possesses strong nucleophilicity (E$_{ox}$ = −1.12 V vs SCE)[37]. Subsequently, radical **I** facilitates the formation of corresponding alkyl radicals **III** via halogen-atom transfer with alkyl iodides. The exothermic dissociation of *α*-iodoamine into iminium iodide provides the thermodynamic energy necessary to drive the process. Next, alkyl radical **III** adds to electron-deficient alkenes to create new C-centered radicals **IV**. These radicals are then reduced at the cathode to form carbanion intermediates **V**, which undergo H$_2$O protonation to generate the final alkylation products **VI**.

In summary, we present a novel electrochemical halogen-atom transfer (*e*-XAT) protocol for the activation of unactivated alkyl iodides using simple, transition-metal-free conditions. Key intermediates in the transformations were shown to be the *α*-aminoalkyl radicals produced by anodic oxidation, which acted as efficient XAT reagents. A wide range of alkyl iodides, including primary, secondary and tertiary, and structurally diverse olefins were successfully converted into corresponding alkylation products, offering access to a diverse array of unnatural amino acid derivatives. The *e*-XAT protocol displayed high functional group tolerance and broad substrate compatibility. With its mild reaction conditions, operational simplicity, and broad functional group compatibility, we anticipate that this innovative *e*-XAT protocol will open a new avenue for future carbon-halogen bond activation and find widespread applications in synthetic and medicinal chemistry.

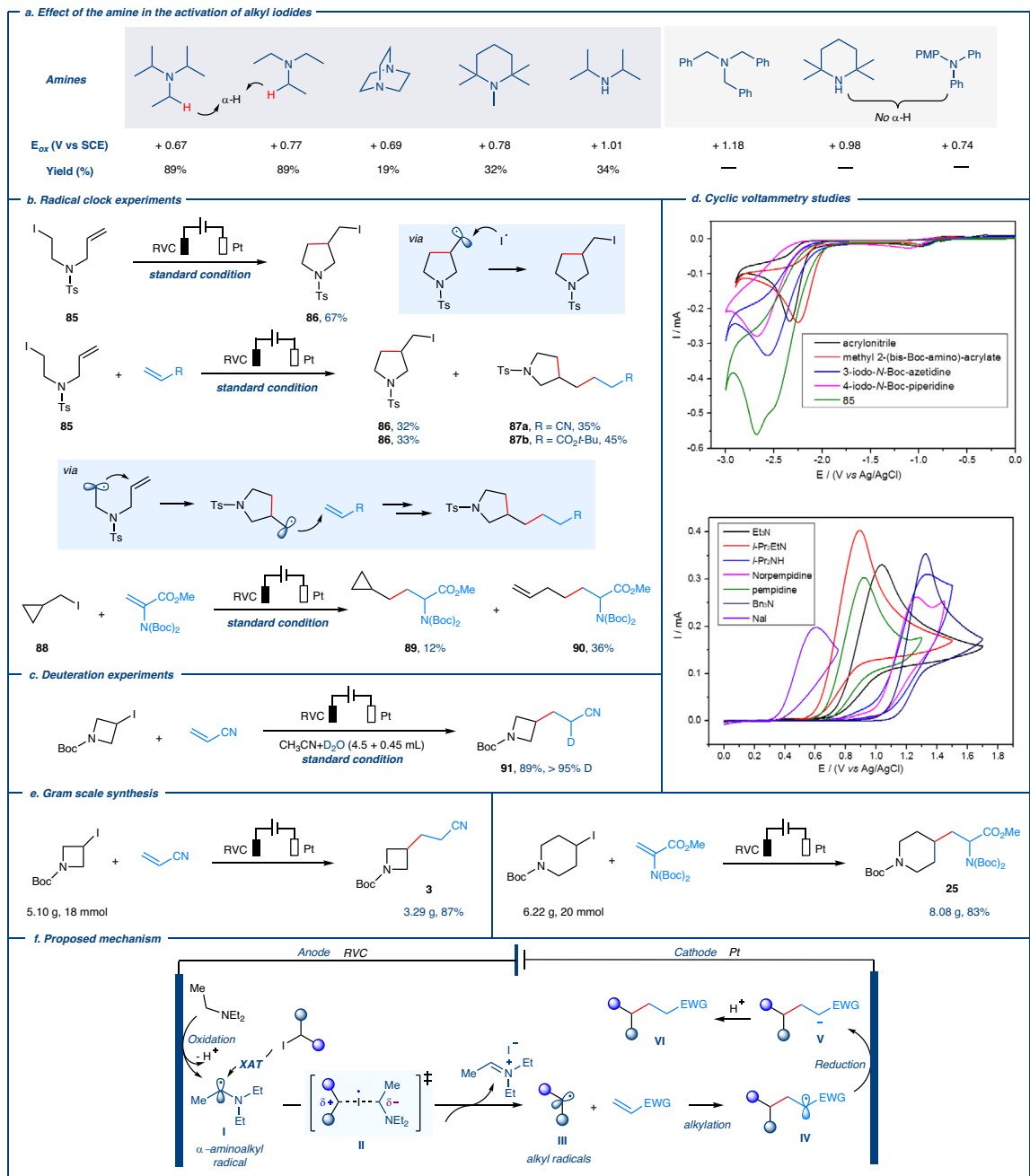

**Fig. 4 | Mechanistic studies and proposed mechanism. a** Effect of the amine in the activation of alkyl iodides. **b** Radical clock experiments. **c** Deuteration experiments. **d** Cyclic voltammetry studies. **e** Gram scale synthesis. **f** Proposed mechanism.

## Methods

### General procedure for the electrochemical halogen-atom transfer (*e*-XAT) alkylation reactions

A Schlenk tube (10 mL) equipped with a magnetic stir bar was charged with the *n*-Bu$_4$NClO$_4$ (102.6 mg, 0.3 mmol, 1.0 equiv), alkyl iodide, if solid (0.3 mmol, 1.0 equiv.), and the alkene, if solid (0.6 mmol, 2.0 equiv.). The tube was equipped with a RVC anode (100 PPI, 1.0 cm × 1.0 cm × 1.2 cm) and a platinum plate (1.0 cm × 1.5 cm) cathode, then sealed, evacuated and refilled with N$_2$ three times. Degassed CH$_3$CN (2.0 mL), Et$_3$N (91.1 mg, 0.9 mmol, 3.0 equiv), distilled water (0.45 mL) and degassed CH$_3$CN (2.5 mL) were sequentially added. The electrolysis was carried out at 50 °C (oil bath temperature) using a constant current of 3.0 mA for 28 h (10 F/mol, secondary alkyl iodide) or 48 h (18 F/mol, primary or tertiary alkyl iodide). Upon completion, the reaction mixture was diluted with EtOAc (30 mL), washed with H$_2$O

(30 mL) and the aqueous layer was extracted with EtOAc (10 mL). The combined organic layers were dried over Na$_2$SO$_4$, filtered and concentrated under reduced pressure. The mixture was purified by flash column chromatography on silica gel.

## Data availability

Materials and methods, experimental procedures, useful information, spectra and mass spectrometry data are available in Supplementary Information. Raw data are available from the corresponding author on request.

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

## Acknowledgements

We thank the Xiaoming Feng laboratory (SCU) for access to equipment. We also thank the comprehensive training platform of the Specialized Laboratory in the College of Chemistry at Sichuan University for compound testing. We acknowledge support from the Ministry of Science and Technology (Nos. 2022YFC2303700 for K.Z.), the National Natural Science Foundation of China (Nos. 22371195 for K.Z.), Sichuan University (Nos. 2020SCUNL204 for K.Z.), and Fundamental Research Funds for the Central Universities.

## Author contributions

K.Z. and X.S. conceived the idea for this work and designed the experiments. X.S. performed and analyzed the experiments. K.Z. supervised the entire project. X.S. and K.Z. discussed the results and wrote the manuscript.

## Competing interests
The authors declare no competing interests.
