## [Peer Review File · Nature Communications]

Electrochemical Halogen-Atom Transfer Alkylation via α -Aminoalkyl Radical Activation of Alkyl IodidesReviewers' Comments:

Reviewer #1:

Remarks to the Author:

In this manuscript Sun and Zheng report an electrochemical method for the hydroalkylation of e-deficient olefins with alkyl iodides via halogen atom transfer (XAT) mediated by aminoalkyl radicals. The virtue of this strategy in comparison with other electrochemical methodologies lies on the fact that it does not need a sacrificial electrode nor reaching deeply negative potentials required to activate alkyl halides via SET. Conceptually, this work builds on a previous report by Leonori and Julia (Science 2020, 367, 1021–1026) describing the same transformation and mechanism, but the present manuscript uses electrochemistry instead of photoredox catalysis to oxidize the amine precursors to generate aminoalkyl radicals. The scope seems quite general and the yields are good, in line with those reported in Leonori's work. The mechanistic proposal is feasible and in agreement with previous works.

This manuscript introduces for the first time XAT with aminoalkyl radicals under electrochemical conditions, which will pave the way to further developments of reactions using this activation mode that are inaccessible via photoredox. This last aspect, together with the generality of the method and the simplicity of the setup (undivided cell, common electrolytes and electrodes), may attract a significant interest of practitioners on the field. Considering these aspects, I would accept this manuscript for its publication in Nature Communications.

Reviewer #2:

Remarks to the Author:

Electrochemical halogen-atom transfer alkylation using triethylamine as a mediator is described. The reactions described in this report are very interesting and probably useful. Especially if alpha aminoalkyl radicals can be used as electrochemical mediators in general, this is a very significant discovery.

However, the discussion of the reaction mechanism in this paper is inadequate. Although the control experiments have successfully demonstrated that this reaction occurs via radical pathway, the initial stage of alkyl radical generation from alkyl iodide should be examined in more detail. The authors cite Reference 37 as the basis for their argument, but the reactions described in this paper are based on photoelectron transfer, and applying the results directly to electrochemical reactions is a somewhat unreasonable argument. In the oxidation of amines and amides, as typified by the Shono oxidation, it is common for iminium ions to be formed by two-electron oxidation and deprotonation, and it is very strange that this process does not occur in the authors' reaction system. I think it is very important to examine the point carefully by conducting the following study:

- 1) Can alpha aminoalkyl radicals be trapped with acrylonitrile? Aminoalkyl radicals can be trapped if the reaction is attempted without alkyl iodides.
- 2) Including the supporting information, only those using RVC electrodes as anodes are listed. Is the electrode material the key to this reaction?
- 3) The reaction also proceeds when 2,2,6,6-tetramethyl-N-methylpiperidine is used. This contradicts their statement that secondary amines are suitable, so further careful verification is needed.
- 4) The mechanism described in Fig 4-f is quite similar to that described in Reference 37. Is it OK?

Dear Reviewers:

We sincerely thank the reviewers for their diligent and meticulous efforts in reviewing our work. The feedback has been invaluable and we appreciate it immensely. We have carefully considered the reviewer's comments and have made the necessary revisions to improve the article. The key revisions and explanations have been outlined below:

For the Reviewer 1

1. "In this manuscript Sun and Zheng report an electrochemical method for the hydroalkylation of e-deficient olefins with alkyl iodides via halogen atom transfer (XAT) mediated by aminoalkyl radicals. The virtue of this strategy in comparison with other electrochemical methodologies lies on the fact that it does not need a sacrificial electrode nor reaching deeply negative potentials required to activate alkyl halides via SET. Conceptually, this work builds on a previous report by Leonori and Julia (Science 2020, 367, 1021–1026) describing the same transformation and mechanism, but the present manuscript uses electrochemistry instead of photoredox catalysis to oxidize the amine precursors to generate aminoalkyl radicals. The scope seems quite general and the yields are good, in line with those reported in Leonori's work. The mechanistic proposal is feasible and in agreement with previous works.

This manuscript introduces for the first time XAT with aminoalkyl radicals under electrochemical conditions, which will pave the way to further developments of reactions using this activation mode that are inaccessible via photoredox. This last aspect, together with the generality of the method and the simplicity of the setup (undivided cell, common electrolytes and electrodes), may attract a significant interest of practitioners on the field. Considering these aspects, I would accept this manuscript for its publication in Nature Communications."

Response: We thank this reviewer for the favorable comments on our work, we really appreciate him/her.

For the Reviewer 2

1. "Electrochemical halogen-atom transfer alkylation using triethylamine as a mediator is described. The reactions described in this report are very interesting and probably useful. Especially if alpha aminoalkyl radicals can be used as electrochemical mediators in general, this is a very significant discovery."

Response: We thank this reviewer for the favorable comments on our work, we really appreciate him/her.

2. "However, the discussion of the reaction mechanism in this paper is inadequate. Although the control experiments have successfully demonstrated that this reaction

occurs via radical pathway, the initial stage of alkyl radical generation from alkyl iodide should be examined in more detail. The authors cite Reference 37 as the basis for their argument, but the reactions described in this paper are based on photoelectron transfer, and applying the results directly to electrochemical reactions is a somewhat unreasonable argument. In the oxidation of amines and amides, as typified by the Shono oxidation, it is common for iminium ions to be formed by two-electron oxidation and deprotonation, and it is very strange that this process does not occur in the authors' reaction system.”

Response: We appreciate the invaluable suggestions provided by this reviewer, which have significantly enhanced the quality of our manuscript. Typically, in the oxidation of amines and amides, as typified by the Shono oxidation, it is common for iminium ions to be formed by two-electron oxidation and deprotonation. However, the electrochemical oxidation process might involve a two-step mechanism in our study. Initially, the alkylamine undergoes a one-electron anodic oxidation, generating the radical cation **II**, which subsequently undergoes deprotonation to yield the α -aminoalkyl radical **III** (as depicted in Scheme 1a). Subsequently, this radical **III** could undergo a secondary oxidation to generate the iminium cation **V** (as illustrated in Scheme 1b). In previous reports and our perspective, the activation of unactivated alkyl iodides with highly nucleophilic α -aminoalkyl radicals was highly fast, and the XAT process was kinetically feasible and exothermic based on density functional theory (DFT) calculations (*Science* **2020**, 367, 1021–1026).

In order to investigate the possibilities if the iminium cation **V** was produced in our *e*-XAT reaction conditions, we conducted a series of control experiments involving TMSCN and CH₃OH as nucleophiles, in combination with three distinct alkylamines—Et₃N, *i*-Pr₂NEt, and 1-methylpiperidine—in the absence of alkyl iodides. The desired nucleophile addition products were not obtained. Additionally, the formation of iminium cation **V** could not be excluded completely due to 3.0 equiv Et₃N was used in our reaction.

Moreover, we conducted an investigation of different electrode materials as anodes and the reaction was completely abolished in the absence of Et₃N (Scheme 3). The results indicated that the triethylamine (Et₃N) was critical for this transformation.

To provide direct evidence of the generation of α -aminoalkyl radicals through anodic oxidation, we conducted radical trapping experiments. These experiments confirmed the successful formation of α -aminoalkyl radicals through the anodic oxidation and deprotonation, as well as the rapidity of the XAT process between α -aminoalkyl radicals and alkyl iodides (for more comprehensive details, please refer to Scheme C3).

The details of these results have been included in the revised SI under section 4.6 titled "Trapping α -aminoalkyl radicals: Exploring the involvement of α -aminoalkyl radicals."

Scheme C1 (Scheme 4.6-1 in SI)

2. “Can alpha aminoalkyl radicals be trapped with acrylonitrile? Aminoalkyl radicals can be trapped if the reaction is attempted without alkyl iodides.”

Response: We appreciate the reviewer's valuable suggestion. We did not observe trapping products of α -aminoalkyl radicals when acrylonitrile and Boc-protected dehydroalanine were introduced as radical acceptors under standard conditions (scheme C2). Our speculation is that the reduction process following the addition of α -aminoalkyl radicals to electron-deficient olefins is highly difficult and could not deliver the desired product through H₂O protonation.

Reaction conditions: Undivided cell, RVC anode (100 PPI), Pt plate cathode, Et₃N (0.9 mmol), alkene (0.6 mmol), CH₃CN (4.5 mL), H₂O (0.45 mL), N₂, 3.0 mA, 50 °C (oil bath temperature), 28 h.

Scheme C2 (Scheme 4.6-2 in SI)

To our delight, the direct evidence comes from the results: In comparison to the alkylation product obtained with a 73% yield in the absence of H₂O under standard conditions for the model reaction (Scheme C3a), the radical trapping products **92-94** were achieved with 72-86% yields when employing 1,4-dicyanobenzene as a radical acceptor (Scheme C3b-c). These results directly demonstrated the successful generation of α -aminoalkyl radicals through anodic oxidation and deprotonation. These radicals are directly captured by the radical anions of 1,4-dicyanobenzene, which are formed via cathodic reduction (*Angew. Chem. Int. Ed.* **2019**, 58, 4058–4062), yielding the cross-coupling products **92-94** (Scheme C3e).

Moreover, the cross-coupling product **95** was achieved with 42% yield when adding alkyl iodide (3-iodo-*N*-Boc-azetidine) to the reaction, arising from the coupling of alkyl radicals with 1,4-dicyanobenzene radical anions. The α -aminoalkyl radical trapping product **92** was not observed (Scheme C3d). These outcomes revealed that upon the generation of the α -aminoalkyl radical, the XAT process ensued immediately, leading to the formation of alkyl radicals.

We have included these experiments and comments in the revised SI under section 4.6 titled "Trapping α -aminoalkyl radicals experiments: involvement of α -aminoalkyl radicals."

Scheme C3 (Scheme 4.6-3 in SI)

3. "Including the supporting information, only those using RVC electrodes as anodes are listed. Is the electrode material the key to this reaction?"

Response: We appreciate the reviewer's constructive suggestions. In response, we have carried out additional experiments with other electrode materials as anodes, such as Carbon felt (CF), Carbon cloth, Carbon rod, Graphite felt (GF), and Pt under standard electrochemical conditions (scheme C4). The results revealed that RVC anode was the best electrode materials and the desired alkylation product **3** was obtained in 89% yield. The use of Carbon felt (CF) and Carbon cloth as anodes led to a slightly drop in reaction efficiency (82% yield). The yield of the desired product was further decreased with using Carbon rod and Graphite felt (GF) as anodes (68% and 67%, respectively). The Pt anode displayed even lower efficacy (42% yield).

Notably, we conducted control experiments with different anode materials in the absence of Et₃N and the reaction was completely abolished, the results showed that triethylamine (Et₃N) was the key for this transformation and not the anode materials. The efficiency of Et₃N was oxidized at different anode materials could be affected the reaction results. So, we think that the Et₃N underwent one-electron oxidation and deprotonation to generate α -aminoalkyl radical was the key for this transformation and the process was highly possible. We have added these experiments in the revised SI (3.1 Reaction optimization: effect of the different electrode materials as anodes).

Scheme C4

4. “The reaction also proceeds when 2,2,6,6-tetramethyl-N-methylpiperidine is used. This contradicts their statement that secondary amines are suitable, so further careful verification is needed.”

Response: We appreciate the reviewer's valuable suggestions. Based on the screening of various amines, we found that the alkylamines with α -hydrogens (α -H; including primary α -H, secondary α -H, and tertiary α -H) and suitable oxidation potentials were capable of generating α -aminoalkyl radicals under electrochemical oxidation conditions, facilitating the transformation. As an example, the amine 2,2,6,6-tetramethyl-N-methylpiperidine with primary α -H could give the alkylation product in reasonable yield (32%). The amines without α -H (unable to generate α -aminoalkyl radicals), such as Ph₂N(PMP) or 2,2,6,6-tetramethylpiperidine, completely suppressed the reactivity.

Control experiments revealed that alkylamines containing secondary α -hydrogens (e.g., Et₃N and *i*-Pr₂NEt) yielded better outcomes compared to the alkylamines containing primary α -hydrogens (like 2,2,6,6-tetramethyl-N-methylpiperidine) or tertiary α -hydrogens (like *i*-Pr₂NEt). This outcome could be attributed to the primary or tertiary α -aminoalkyl radicals were more difficult to generate during the initiating stage (electrochemical oxidation stage), while secondary α -aminoalkyl radicals demonstrated better stability (compared to primary α -aminoalkyl radicals) and less steric hindrance (compared to tertiary α -aminoalkyl radicals) in the XAT step.

These insightful observations have been added into the revised manuscript, highlighted with a yellow background.

5. “The mechanism described in Fig 4-f is quite similar to that described in Reference 37. Is it OK?”

Response: We appreciate the reviewer's insightful question. Based on a series of mechanistic experiments and the previous reports, we think that the proposed mechanism for the *e*-XAT strategy illustrated in Figure 4f was rational and the most possible. First, the process of electrochemical oxidation alkylamine could be involved in two steps, we consider that the alkylamine initially undergoes one-electron anodic oxidation and subsequent deprotonation, leading to the formation of the crucial α -

aminoalkyl radical. We conducted an investigation of different electrode materials as anodes and the reaction was completely abolished in the absence of Et₃N. The results indicate that triethylamine (Et₃N) was critical for this transformation.

Furthermore, the control experiments by using TMSCN and CH₃OH as nucleophiles to trap iminium cations generated through a second oxidation step without the presence of alkyl iodides in our e-XAT reaction system. No desired nucleophile addition products were observed yet. Moreover, the radical trapping experiments directly demonstrated that the α -aminoalkyl radicals were successfully generated by the anodic oxidation, deprotonation and the XAT process between α -aminoalkyl radicals and alkyl iodides was very fast (more detail see Scheme C3). Considering the previous reports, the activation of unactivated alkyl iodides with highly nucleophilic α -aminoalkyl radicals was highly fast, and the XAT process was kinetically feasible and exothermic based on density functional theory (DFT) calculations. So, as depicted in Figure 4f, we considered that the alkylamine undergoes one-electron anodic oxidation followed by deprotonation, ultimately generating the α -aminoalkyl radical. This α -aminoalkyl radical acts as an efficient XAT reagent, facilitating the formation of alkyl radicals through halogen-atom transfer with alkyl iodides.

In conclusion, based on above evidences, we considered that the proposed mechanism depicted in Figure 4f for the e-XAT strategy is reasonable and highly possible.

Reviewers' Comments:

Reviewer #2:

Remarks to the Author:

I believe that the revised manuscript has been properly and very carefully revised, including additional experiments, and that all the questions I commented on have been clarified. For these reasons, I consider this paper acceptable for publication.

RE: *Nature Communications*

Manuscript number: NCOMMS-23-22251A

Manuscript Title: "Electrochemical Halogen-Atom Transfer Alkylation via α -Aminoalkyl Radical Activation of Alkyl Iodides"

Author(s): Xiang Sun, Ke Zheng

We want to express my sincere gratitude for the time and effort the reviewers dedicated to reviewing our article. Reviewer's constructive feedback has been invaluable and we appreciate it immensely. We have carefully edited the manuscript to comply with the policies and formatting requirements of the journal.

For the Reviewer 2

I believe that the revised manuscript has been properly and very carefully revised, including additional experiments, and that all the questions I commented on have been clarified. For these reasons, I consider this paper acceptable for publication.

Response: We thank this reviewer for the favorable comments on our work, we really appreciate him/her.

We sincerely thank you for your diligent and meticulous efforts in reviewing our work. Your feedback is highly valued. We are resolutely committed to enhancing the manuscript to meet the high standards of "*Nature Communications*". We hope this revised version meets the requirements for publication in this esteemed journal.